# The Role of Oxidative Stress and Antioxidants in Cardiovascular Comorbidities in COPD

**DOI:** 10.3390/antiox12061196

**Published:** 2023-05-31

**Authors:** Zsuzsanna Miklós, Ildikó Horváth

**Affiliations:** 1National Korányi Institute for Pulmonology, Korányi F. Street 1, H-1121 Budapest, Hungary; miklos.zsuzsanna@koranyi.hu; 2Department of Pulmonology, University of Debrecen, Nagyerdei krt 98, H-4032 Debrecen, Hungary

**Keywords:** antioxidants, arterial aging, airway inflammation, α-Klotho, ROS, hydrogen peroxide, SOD, heart rate variability

## Abstract

Oxidative stress driven by several environmental and local airway factors associated with chronic obstructive bronchiolitis, a hallmark feature of COPD, plays a crucial role in disease pathomechanisms. Unbalance between oxidants and antioxidant defense mechanisms amplifies the local inflammatory processes, worsens cardiovascular health, and contributes to COPD-related cardiovascular dysfunctions and mortality. The current review summarizes recent developments in our understanding of different mechanisms contributing to oxidative stress and its countermeasures, with special attention to those that link local and systemic processes. Major regulatory mechanisms orchestrating these pathways are also introduced, with some suggestions for further research in the field.

## 1. Introduction

Oxidative stress driven by several environmental and local airway factors associated with chronic obstructive bronchiolitis, a hallmark feature of chronic obstructive pulmonary disease (COPD), plays a crucial role in the disease pathomechanisms [1,2]. Unbalance between oxidants and antioxidant defense mechanisms amplifies the local inflammatory processes, has systemic effects, contributes to developing COPD–related comorbidities, and worsens cardiovascular health. COPD often coexists with cardiovascular diseases (CVDs). CVDs are not only the most common comorbidities perceived in COP, but also account for an increased risk of death in COPD patients [3,4,5]. Approximately 30% of COPD patients are reported to die due to CVD. COPD and CVD share common pathophysiological mechanisms strongly related to oxidative stress [6]. This review summarizes our current understanding of the local and systemic processes that link COPD and various CVDs via oxidative stress. We focus on some relevant mechanisms that orchestrate the systemic responses leading to the parallel development of respiratory and cardiovascular dysfunctions. We aim to update and extend previous reviews related to the field by describing biomarkers, discussing the relationship between COPD and CVDs in a broader sense instead of focusing on certain specific CVDs and highlighting novel, less investigated mechanisms connecting the two disease entities via oxidative stress [4,7,8,9,10,11].

## 2. Pathways of Oxidative Stress

Oxidative stress is when the oxidative burden imposed by exposure to exogenous and endogenous free radicals exceeds the antioxidant defense capacities. This may occur due to excessive oxidant production, exhaustion, or the defective functioning of antioxidant mechanisms (Figure 1). Reactive oxygen species (ROS), such as hydroxyl radical and superoxide anion, are produced by each cell in the body during mitochondrial respiration and cell signaling processes. ROS production by the immune, mainly phagocytic cells, is also essential in the immune defense against pathogens [1,2]. To protect the physiological function of cells from the harmful effects of exogenous and endogenous radicals, the body maintains powerful antioxidant mechanisms.

### 2.1. Production of Oxygen Radicals

Phagocyte ROS generation relies on the operation of nicotinamide adenine dinucleotide phosphate (NADPH)-oxidase (NOX) enzymes which produce superoxide anion (O_2_^•−^) by transferring an electron from NADPH to O_2_ as a result of activation of nuclear factor kappa-light-chain-enhancer of activated B cells (NF-κB) signaling. NOX enzymes are localized to the membrane, and their different isoforms are expressed in numerous tissues and cell types in the body [12,13]. The O_2_^•−^ anion is unstable and is rapidly dismutated to hydrogen peroxide (H_2_O_2_) by the enzyme superoxide dismutase (SOD) [14]. Phagocyte lysosomes also contain the enzyme myeloperoxidase, which catalyzes the conversion of H_2_O_2_ to hypochlorous acid (HOCl), a highly oxidizing agent [15]. H_2_O_2_ can also be converted to reactive nitrogen and carbonyl species (RNS and RCS) in the Haber–Weiss and Fenton reactions [16,17].

A further essential source of ROS is excessive nitrogen monoxide (NO) production by inducible nitric oxide synthase in phagocytes and various cell types as part of the inflammatory responses [18]. When NO and O_2_^•−^ are present at increased concentrations, as seen in inflammation, they readily combine to form peroxynitrite (ONOO^−^). Peroxynitrite is a highly reactive oxidant with enhanced stability [2,18].

Reactive species can oxidize thiols, amines, and amino acid residues of proteins such as cysteine, methionine, and tyrosine. This may alter the tertiary structure and function of the protein. In addition, ROS can also be harmful to lipids and DNA, which may cause membrane dysfunction and transcriptional errors [1,2,19,20,21].

Nuclear factor-κB (NF-κB) signaling connects ROS production to local and systemic inflammation in various diseases. While certain NF-κB regulated genes control ROS generation by the cell, ROS also have complex inhibitory and stimulatory effects on NF-κB signaling, mediating mainly proinflammatory responses [22]. Another transcription factor that is relevant in exacerbating ROS production in inflammatory responses is activator protein 1 (AP-1). AP-1 activity is redox-sensitive and induced by many physiological, pathophysiological, and environmental stimuli, including various cytokines and bacterial and viral infections [23]. The products of AP-1-induced genes participate in inflammatory processes and ROS production and have been shown to contribute to the etiology of disease conditions in the respiratory and cardiovascular systems [23,24,25].

### 2.2. Antioxidative Defense

The action of ROS is kept under control by enzymatic and non-enzymatic defense mechanisms [1,2]. Antioxidant molecules, metal-binding proteins, and unsaturated lipids acting as electron donors or recipients can scavenge non-enzymatic radicals. In the lung, the antioxidants vitamin C (ascorbate) and vitamin E (tocopherol) are found in abundance in the airway surface liquid [26,27]. In addition, albumin, mucin in extracellular body fluids and glutathione within cells are relevant scavengers as they offer methionine and cysteine residues for radicals [28,29,30].

Enzymatic ROS antioxidation is carried out by three significant enzymes, superoxide dismutase (SOD), catalase and glutathione peroxidase (GPx). Superoxide dismutase (SOD1, SOD2 and SOD3) quickly remove O_2_− by converting it to H_2_O_2_ to prevent it from causing damage or producing extremely damaging peroxyl radicals [14]. However, this process produces H_2_O_2,_ which can be the precursor of further hydroxyl radical generation. Catalase and GPx eliminate H_2_O_2_ by splitting it into H_2_O and O_2_ [31,32]. In the GPx-catalyzed reaction, glutathione (GSH) acts as a hydrogen ion donor, becoming glutathione disulphide (GSSG). Expression of antioxidant enzymes is highly regulated by the transcription factor ‘nuclear factor erythroid 2-related factor 2 (Nrf2)’. Decreased activation of Nrf2 due to inflammatory cytokines and depression of anti-aging mechanisms participates in the downregulation and loss of antioxidant defense in COPD and CVDs [33,34,35,36]. Moreover, Nrf2 is downregulated by oxidative stress itself, initiating a vicious circle [37,38,39].

### 2.3. Sources of Oxidative Stress in COPD

In COPD development, exogenous radicals from cigarette and biomass smoke, air pollution, and occupational exposure contribute substantially to the oxidative stress of small molecules [2,40,41]. In addition, cigarette smoke can enhance NOX activity in lung tissue and stimulate leukocyte migration [42,43]. Compared with non-smokers, the neutrophil count in COPD patients is higher in BAL fluid and sputum and enhanced NOX activity can be detected in circulating neutrophils [12,43]. Moreover, NOX4 was upregulated in COPD patients’ airway smooth muscle cells, correlated with disease severity, and was associated with pulmonary hypertension [43,44,45,46].

Furthermore, the increased oxidant burden causes the upregulation of antioxidant genes that play protective roles. For example, the induction of the GSH gene increases the accumulation of GSH in the epithelial lining fluid in the airspaces, which is important for preventing oxidative injury [47,48]. Similarly, increased SOD and catalase activity have been observed in the sputum of COPD patients during acute exacerbation [49]. On the other hand, cigarette smoke exposure and long-term inflammation have been shown to reduce the activity of antioxidant enzymes, such as catalase and superoxide dismutase, contributing to the severe perturbation of oxidative balance in the lung tissue [50,51] (see in details later).

## 3. Oxidative Stress—A Link between COPD and Cardiovascular Comorbidities

COPD and CVDs share common pathophysiological mechanisms that involve systemic inflammation, endothelial dysfunction, vascular inflammation and remodeling, alteration in heart rate variability, and clotting abnormalities [6]. These underlying mechanisms (at least in part) participate in the development of pulmonary arterial hypertension (PAH), hypertension, accelerated atherosclerosis and its consequences, such as stroke, ischemic heart disease and, in the long run, cardiac failure (Figure 2).

### 3.1. COPD and Vascular Aging, Hypertension

Though widely debated, many experts view the development of COPD as a manifestation of accelerated aging [52]. Indeed, a strong association between vascular aging and COPD is well-established in the literature. COPD manifests as small airway obstruction (chronic obstructive bronchiolitis) and emphysema. Pathologically, chronic inflammation and fibrosis of peripheral airways, increased mucus secretion, luminal accumulation, and destruction of lung parenchyma and alveoli are typical alterations. These overlapping phenotypes may manifest with varying severity and might dominate the clinical picture of individual patients [52,53]. The aging vasculature is characterized by fibrotic remodeling and thickening of the arterial wall, intima-media hyperplasia, and endothelial dysfunction [39,54,55,56]. The aging arteries stiffen, and the consequential alteration in their biomechanical properties is a critical factor in developing hypertension, one of the significant CV comorbidities in COPD [54,57]. Early vascular aging is best detected by measuring pulse wave velocity (PWV), as pulse propagation is typically faster in stiffer, aged arteries. Numerous studies have found that PWV is abnormally high in COPD patients [58,59]. Arterial stiffness, as measured by PWV, was independently associated with the severity of emphysema [60] and airway obstruction [59,61,62,63,64]. Furthermore, it was established by studying twins that the link between lung function and arterial stiffness is not genetically determined. However, there is a phenotypic association between spirometric parameters used to assess airway obstruction, such as forced vital capacity (FVC) and forced expiratory volume in 1 s (FEV_1_), and augmentation index, a marker of pulse wave reflection pointing towards shared pathways of their co-development in COPD [65]. The observations that arterial stiffness seems more severe in frequently exacerbating COPD patients and to intensify acutely during exacerbation suggest a dynamic, reversible component of this relationship that is not fully characterized [66]. COPD rehabilitation programs have been shown to benefit arterial stiffness in a subpopulation of patients significantly but not in general [67]. This observation is similar to those demonstrating that lung function values or even regulatory molecules known as part of antioxidant defense cannot be improved much by these programs despite their well-documented positive effects on the overall health status of involved patients [68]. It is also worth mentioning that COPD often associates with obstructive sleep apnea (OSA) [69]. OSA is widely recognized as a significant risk factor for developing arterial hypertension and its complications [70]. Among the underlying mechanisms, the contribution of hypoxic periods during sleep in OSA to oxidative stress has the utmost relevance [71].

Among the common underlying mechanisms of vascular aging and COPD, persistent systemic low-grade inflammation, oxidative stress (i.e., overproduction of reactive oxygen species and decreased antioxidant capacity) and deterioration of anti-aging mechanisms have critical relevance.

#### 3.1.1. Oxidative Stress in COPD and Vascular Aging

Enhanced oxidative stress plays a significant role in COPD and vascular aging pathogenesis. It is attributable to various pathophysiological mechanisms involving mitochondrial senescence, NADPH oxidase (NOX) overactivation, endothelial dysfunction, overactivation of the tissue renin-angiotensin-aldosterone system (RAAS), and also to COPD-related hypoxia [1,2,39,54,55,56,57,72].

The sources of oxidative stress are manifold in both conditions. Cigarette smoke exposure, a significant risk factor in COPD and vascular aging, is a direct source of inhaled oxidants and irritants that generate inflammation. In COPD, dysfunctional mitochondria of structural cells (airway epithelium, fibroblasts), NADPH oxidases (NOX) of airway epithelial cells, and myeloperoxidase enzymes of neutrophils and macrophages produce a substantial amount of reactive oxygen species [1,2,26,52]. An increased ROS production by mitochondria and NOX enzymes is also typical in the aging vasculature [54,73]. Inflammatory cytokines, adipokines, activation of endothelin1 and angiotensin1 receptors, and dysfunctional NO synthase operation further aggravate oxidative stress by activating NOX enzymes both in the vasculature and the lung [2,26,36,55,57,74,75].

Oxidative stress is further amplified by the decreased antioxidant capacity of the lung and vascular tissue [39,55,56,75,76,77]. Reduced superoxide dismutase (SOD) activity has been observed in relation to vascular aging [57,75,78]. Though acute exacerbations in COPD are associated with increased extracellular SOD activity [49], altered SOD function due to SOD2 and SOD3 gene polymorphism has been implicated in the etiology of COPD [79,80]. Lower antioxidant capacity is also reflected by lower circulating and cellular glutathione concentrations in COPD and during vascular aging [39,81]. However, glutathione concentrations measured in BAL fluid and sputum are elevated in COPD [82]. In addition, the transcription factor Nrf2 is downregulated and exhibits impaired activation in response to oxidative stress [37,38,39]. This results in decreased expression of several antioxidant enzymes in the lung and vascular tissue [33,34,35,36] and ROS production by NOX in the vasculature [37]. Catalase activity is reduced in COPD patients [83,84], and a decreased expression was found in the bronchial epithelium [50].

In contrast, during acute exacerbation, enhanced catalase activity can be observed in the sputum [49]. Decreased catalase activity has been linked to several age-related diseases, including cardiovascular disorders [32]. Concerning GPx activity, a decrease was observed in erythrocytes [83,85,86,87], and blood and plasma samples [88,89,90] of COPD sufferers. Aging and vascular abnormalities have also been related to depressed GPx functioning by several studies [91,92,93]. Nitrative stress is also well-documented in COPD and is further aggravated during exacerbations [94]. Defected heme-oxygenase-1 (HO-1) signaling also contributes to decreased antioxidant and anti-inflammatory defense in lung and cardiovascular diseases. HO-1 is an inducible stress protein implicated in chronic airway inflammation [95]. The major activity of HO-1 is to eliminate the high oxidant-free heme by converting it to biliverdin, ferrous iron and carbon monoxide. Its expression is strongly influenced by Nrf2 [95].

#### 3.1.2. The Consequences and Aggravators of Oxidative Stress

The consequences of oxidative stress include inflammation, disruption of anti-aging processes and endothelial injury, which typically manifest in a systemic form in COPD. Though oxidative burden is a key factor in igniting these processes, they also fuel and aggravate oxidative stress by activating signaling pathways that induce ROS production and/or downregulate antioxidant defense mechanisms.

**Systemic inflammation.** Oxidative stress induces redox-sensitive proinflammatory signaling in various cell types. Increased generation of ROS species is associated with the activation of proinflammatory transcription factors and proteins such as NF-κB, activator protein 1, transforming growth factor-β (TGF-β), different isoforms of matrix metalloproteinases, p38MAPK both in the lung and vascular tissue. Activation of these pathways results in the upregulation and release of inflammatory cytokines (i.e., TGF-β, TNF-α, IL-1, IL-6), chemokines and adhesion molecules that perpetuate inflammation locally and systemically [2,35,36,52,55,57,73,74,75,96]. In addition, local inflammation triggers maladaptive remodeling. Activation of MMPs breaks down elastic fibers, and profibrotic processes (activation of local RAAS and fibroblasts) operate to give rise to small airway fibrosis and emphysema in the lung, and intima-media thickening and calcification in the arterial wall [39,52,54,57,58,72,97].

**Endothelial abnormalities.** Endothelial injury and dysfunction are also obligate consequences of long-term oxidative stress in the lung and vasculature and common features of COPD and arterial aging [96]. The normal endothelium releases. NO, is a gaseous signaling molecule which has beneficial effects on systemic and pulmonary vasculature. It decreases vascular tone, has an antiproliferative impact on smooth muscle cells, and inhibits platelet aggregation and the release of inflammatory mediators. In oxidative stress, superoxide species react with NO to form peroxynitrite, a short-lived, highly potent oxidant that induces cell injury and mediates proinflammatory processes [18]. In addition, in oxidative stress, tetrahydrobiopterin, a cofactor of NO synthase (NOS), gets oxidized leading to NOS uncoupling. The uncoupled NOS produces superoxide instead of NO, further exacerbating oxidative stress. As a result, the bioavailability of NO decreases and its beneficial effects deteriorate [98,99].

Furthermore, NOS activity is reduced due to the accumulation of its endogenous inhibitor, asymmetric dimethylarginine (ADMA) [100]. Elevated plasma ADMA levels have been associated with endothelial dysfunctions and cardiovascular diseases, including ischemic stroke, pulmonary hypertension, and heart failure [101]. In addition, the bioavailability of NO is further aggravated by the upregulation of arginase, the enzyme that cleaves l-arginine, the precursor of NO [102]. Elevation of arginase activity reduces the availability of l-arginine to NOS, which can reduce NO formation, uncouple NOS, and increase peroxynitrite production contributing to airway hypercontractility and vascular remodeling [100,103,104,105]. Moreover, NOS expression and activity are directly reduced by cigarette smoke exposure, oxidative stress, and inflammatory processes [99,106,107]. Besides uncoupled eNOS, activation of xanthine oxidase and NADH/NADPH oxidase pathways by ROS and RNS contained in cigarette smoke and generated by inflammatory cells makes endothelial cells an important source of further ROS production [18].

Oxidative stress contributes to endothelial dysfunction also by inducing increases in lipid peroxidation [108,109] and AGE-RAGE activation [110]. In addition, decreased antioxidant capacity in the lung tissue (Nrf2 downregulation in epithelial cells [111,112]), the direct toxic effect of cigarette smoke exposure (by stimulating endothelial cell apoptosis) [113,114], and endothelial cell senescence induced by oxidative stress and smoking also may play a role in the pathogenesis of endothelial dysfunction [96].

As a result of endothelial derangement, proliferative and fibrotic processes dominate vascular homeostasis and vascular contractility increases. Endothelial injury has been reported to affect the etiology of various COPD-related vascular disorders, such as pulmonary arterial hypertension, hypertension, renal dysfunction, and venous thromboembolism [96,99]. The damaged endothelium is a critical factor in developing CVD complications and promotes the progression of emphysema. Several human and animal model studies provided evidence for a link between endothelial damage and emphysema [96,115]. Moreover, a model study with rats showed that treatment with vascular endothelial growth factor (VEGF—a trophic factor promoting endothelial cell survival) inhibitors initiated emphysema development without inflammation [116]. However, stimulators of soluble guanylate cyclase (a target enzyme of NO in smooth muscle cells) in a rodent model exposed to cigarette smoke were beneficial for pulmonary vascular remodeling and prevented emphysema progression [117]. Another potential link between emphysema and endothelial dysfunction in COPD might be the aberrant purinergic signaling and elevated pulmonary ATP levels with plausible interactions with ongoing oxidative stress [118,119,120].

**Accelerated aging.** Oxidative stress contributes to the development of COPD and related CV disorders by weakening and disrupting certain anti-aging processes, such as sirtuin activity and balance of the Klotho protein—fibroblast growth factor (FGF) 23 system, and also by aggravating processes that stimulate cellular senescence, such as telomere shortening and adverse epigenetic modifications. Sirtuins (SIRTs) are enzymes of the silent information regulator 2 (Sir2) class III deacetylase family. As their activity is regulated by NAD^+^, they are highly redox-sensitive. They participate in biological processes, which include cellular response mechanisms against a wide range of stressors. SIRTs modulate transcription, cell growth, oxidative stress-tolerance and metabolism and thereby help to alleviate aging-related mitochondrial dysfunction, genomic instability, and inflammation [121,122]. Among the seven mammalian sirtuins, SIRT1 and SIRT6 have been implicated to have protective effects against COPD. SIRT1 and SIRT6 are downregulated by cigarette smoke exposure and in the lungs of COPD patients [123,124,125]. SIRT1 is known to deactivate redox-sensitive transcription factor NF-κB by deacetylating its RelA/p65 subunit [126]. NF-κB stimulates the transcription of proinflammatory genes (e.g., IL-8, IL6, TNFα) [126]. Therefore, reduced levels of SIRT1 enhance the proinflammatory effects of oxidative stress and contribute to the pathogenesis of COPD. Lower SIRT1 activity may participate in COPD development by promoting senescence in different cell types of the lung tissue, as SIRT1 is also known to deacetylate p53 and negatively regulate the forkhead box O3 (FOXO3) pathway that is involved in the transcription of genes responsible for cellular senescence [127,128]. SIRT6 has also been shown to have effects which may be protective against COPD by antagonizing the senescence of human bronchial epithelial cells [129].

Impaired sirtuin activity also plays a crucial role in aging-associated vascular remodeling [39,55,56,130]. SIRT1 is highly expressed in endothelial cells, and it directly activates eNOS in the cytoplasm and increases eNOS expression. By inhibiting p53, forkhead box O1 (FOXO1) [131], and plasminogen activator inhibitor-1 pathways [132], it protects against endothelial senescence. Acting in vascular smooth muscle cells inhibits migration and proliferation, tunica media remodeling, and protects against DNA damage, neointima formation and atherosclerosis [133,134,135]. SIRT6 inhibits proprotein convertase subtilisin/kexin type 9 (PCSK9) and insulin-like growth factor (IGF)-Akt signaling in the vasculature, thereby reducing senescence and protecting against vascular aging [39,130].

The FGF23—α Klotho (KL) system has emerged as an endocrine axis essential for maintaining phosphate homeostasis. FGF23 is a bone-derived hormone, and its binding to its FGF receptor in the kidney and parathyroid gland requires KL as an obligate co-receptor [136]. KL is a transmembrane protein, but it also occurs in a soluble form in the blood produced by either alternative splicing or proteolytic cleavage [137,138]. KL has been attributed to anti-inflammatory and anti-senescence effects [136]. In addition, the Klotho protein protects cells and tissues from oxidative stress. The mechanisms include activating FOXO transcription factors and the NF-κB and Nrf2 pathways [139,140,141]. Transgenic mice are deficient in Klotho exhibit phosphate retention, accelerated aging, and lung emphysema [142]. Therefore, it has been postulated that Klotho is protective against COPD development. Despite this, studies investigating the association between KL and COPD are scarce in the literature, and the findings are controversial. Gao et al. found that KL expression was decreased in the lungs of smokers and further reduced in patients with COPD [139].

Moreover, they found that KL depletion increased cell sensitivity to cigarette smoke-induced inflammation and oxidative stress-induced cell damage in a mouse model. In the blood, a slightly lower KL level was measured by Patel et al. in COPD patients [143], while Pako et al. detected decreased KL levels in OSA [144]. However, other studies found that plasma KL levels did not correlate with clinical parameters in stable COPD patients [145], and their levels were not affected by pulmonary rehabilitation [68].

The FGF 23—KL axis has also been shown to be associated with cardiovascular health [146]. Several studies have found an inverse relationship between KL concentrations and the likelihood of having CVD [147,148]. Arking et al. identified a KL gene variant (KL-VS) which conferred cardioprotective advantages on heterozygous subjects concerning high-density lipoprotein cholesterol levels, systolic blood pressure, stroke, and longevity. Interestingly, they found that homozygosity for KL-VS is disadvantageous compared to wild-type genetic background [149]. Using mouse models, Hu et al. proved an association between KL levels and vascular calcification. They found that overexpression was protective, whereas KL deficiency promoted calcium deposition in the vessel wall [150]. KL deficiency was also found to participate in the development of salt-sensitive hypertension through vascular non-canonical Wnt5a/RhoA activation [151]. The significant cardioprotective effect of KL may be the suppression of inflammation and oxidative stress in vascular smooth muscle (VSMC) and endothelial cells. KL inhibits phosphate entry in VSMCs through the PiT1 carrier, which is known to stimulate the production of ROS. In addition, KL inhibits sodium overload-induced ROS production in endothelial cells [152].

Telomere shortening and epigenetic modifications of the DNA are hallmarks of aging, and both are accelerated by oxidative stress [153,154]. Oxidative stress and inflammation influence the cell’s epigenetic machinery, from DNA and histones to histone modifiers resulting in adverse modifications, such as hydroxylation of pyrimidines and impaired DNA demethylation [154]. Enhanced tissue and leukocyte telomere shortening and various epigenetic modifications be associated with the development of COPD and vascular remodeling [56,155,156,157].

**Alfa-1 antitrypsin deficiency.** Alfa-1 antitrypsin deficiency (A1ATD) is a hereditary disease that is the consequence of the genetic mutations of the SERPINA1 gene and predisposes homozygous and heterozygous subjects to the development of emphysema and liver disease. Although it is considered a rare disease, several authors have proposed that it might not be rare but severely underdiagnosed [158]. The genetic disorder leads to the accumulation of misfolded proteins in α1-antitrypsin producing cells, mainly in hepatocytes and, to a lesser extent, in lung epithelial cells. The main function of alfa-1 antitrypsin is to antagonize neutrophil elastase activity, but it also operates as an acute phase protein with anti-inflammatory effects. In its absence, the degradation of elastin fibers and extracellular tissue matrix in the lung overactivates upon activation of neutrophil cells and promotes the development and progression of emphysema [159,160]. In addition, the additive effect of cigarette smoke exposure multiplies the risk of emphysema.

The effect of A1ATD on the cardiovascular system is also manifold but controversial. The degradation of elastic elements in the vessel wall impairs its physiological distensibility. As a result, arterial compliance increases and the Windkessel function gets compromised. A recent study of 91,353 subjects has shown that this decreases systolic and diastolic blood pressure values [161]. Losing elastic properties can also lead to aorta distension and aneurysms [162,163]. In addition, the absence or lower level of alfa-1 antitrypsin is associated with inflammatory vascular diseases such as fibromuscular dysplasia and ANCA-positive vasculitis [164,165].

Several studies in animal models and humans indicate that AA1TD is associated with enhanced oxidative stress and decreased antioxidant defense even at early stages of disease progression [166,167,168,169].

### 3.2. COPD and Pulmonary Arterial Hypertension (PAH)

Pulmonary arterial hypertension and consequential right heart failure are common cardiovascular complications in COPD. The prevalence of PAH is 5% in moderate (GOLD stage II), 27% in severe (GOLD stage III), and 53% in very severe (GOLD stage IV) COPD [170]. As the diagnostic criterion for PAH is mean pulmonary arterial pressure ≥25 mmHg at rest, these statistics reflect an advanced stage of pulmonary circulation abnormality. Several studies on animal models as well as human studies, however, have shown that pulmonary vascular changes occur in mild COP, or even before the development of lung emphysema [171,172,173]. Moreover, right ventricular dysfunction and remodeling have been observed in COPD patients without PAH [174,175].

Vascular changes in COPD are characterized by remodeling the pulmonary vessels and endothelial dysfunction [176,177]. In addition, vascular derangement in emphysema may also contribute to the pathogenesis of PAH [176]. Pulmonary arterial remodeling affects mainly the intimal layer. Intimal hyperplasia develops due to the proliferation of poorly differentiated smooth muscle cells and extracellular matrix deposition [9,178]. In addition, pulmonary arterial stiffening increases right ventricular afterload and the pulsatile load on the pulmonary microcirculation [177]. The latter induces endothelial dysfunction and inflammation in the distal pulmonary vasculature [179,180].

Pulmonary endothelial dysfunction is an early injury in PAH development and has similar mechanisms and consequences as in systemic circulation (see above). It is characterized by reduced expression of eNOS, diminished production of NO and prostacyclin, increased secretion of endothelin, and expression of TGFβ receptors. These alterations promote vasoconstriction and contribute to pulmonary vascular remodeling.

Several underlying factors have been identified that precipitate vascular changes in COPD-related PAH, such as hypoxia, activation of sympathetic nerves, cigarette smoking, biomass smoke exposure, and epithelial cell injury [176]. Hypoxia is a well-established cause of pulmonary vascular remodeling and PAH. However, its role in COPD-related PAH is debated, as vascular abnormalities are present even in patients with mild COPD and without hypoxemia [176]. Acting on smooth muscle cells, endothelial cells and fibroblasts, hypoxia can induce cell proliferation by inhibiting antimitogenic and stimulating mitogenic stimuli and increasing the production of inflammatory mediators. [181] A key factor linking hypoxia to the activation of these pathways and oxidative stress is the hypoxia-inducible factor 1 (HIF-1) [182], the serum level of which is elevated in COPD patients [183,184]. COPD is also associated with increased sympathetic tone and activation of the renin-angiotensin-aldosterone system. This neurohormonal imbalance favors increased oxidative stress and activation of inflammatory and fibrogenic responses, which lead to adverse remodeling in the heart and vasculature [185]. Cigarette smoke and biomass smoke stimulate vascular remodeling by direct toxic effects on the endothelial cells by enhancing gene expression and release of inflammatory cytokines locally and systemically [186,187], downregulating eNOS [188] and inducing oxidative and nitrative stress [2,96,176]. In addition, injured bronchial epithelial cells in COPD are considered to orchestrate many immune and inflammatory processes in COPD pathogenesis, also contributing to vascular remodeling [189,190].

### 3.3. COPD and Accelerated Atherosclerosis

Atherosclerosis is the leading cause of stroke, coronary heart disease and peripheral arterial disease, which are responsible for a high percentage of mortality in COPD patients. COPD and atherosclerosis share several common risk factors and underlying mechanisms, such as cigarette smoking, sedentary lifestyle, oxidative stress, endothelial dysfunction, high blood pressure and adverse platelet activation [4,11]. In addition, several studies indicate that the severity of COPD and airflow limitation correlate with the severity of atherosclerotic disease [191,192].

Indeed, several pathophysiological mechanisms observed in COPD participate in the progression of atherosclerosis [10]. The impaired endothelial function has relevance at the early stages of plaque formation, as the inflammatory profile of the injured endothelium enhances the secretion of adhesion molecules, increases the permeability of the endothelial barrier, and aids the recruitment of inflammatory immune cells to the lesion [193]. In addition, systemic inflammation and increased oxidative stress can fuel plaque development by aggravating local inflammatory processes in vulnerable sites of the arterial tree and promoting the oxidization of low-density lipoprotein particles [10,194,195,196].

### 3.4. COPD and Cardiac Diseases

COPD often associates with various abnormalities of cardiac function that lead to heart failure (HF). The prevalence of HF in COPD ranges from 7–42% [8]. The effect of PAH on right ventricular function is well documented. The increased afterload of the right heart initiates maladaptive remodeling processes, and right heart failure develops [177,197]. The early signs of right ventricular dysfunction begin to develop at the early stages of PAH progression, even when pulmonary arterial pressures are in the normal range, but signs of pulmonary vascular derangement are already present [174,175,198]. COPD exacerbations impose an additional load on the heart due to hypoxic pulmonary vasoconstriction and hyperinflation of the lung [199,200]. Maladaptive alteration in the right heart also led to dilatation and electrical remodeling of the right atrium and ventricle, which increases the risk of cardiac arrhythmias [197,201].

Abnormal lung function in COPD also affects the function of the left heart. Emphysema-related hyperinflation of the lung and depressed right ventricular function impairs left ventricular filling and reduces cardiac output [197,202]. Hypoxemia observed in more severe COPD and during exacerbations can increase the risk of cardiac ischemia, and due to altered repolarization, the risk of ventricular arrhythmias and sudden cardiac death [199,201,203]. In addition, cardiac ischemia exposes the heart to oxidative stress that causes derangements in cardiomyocyte homeostasis, such as disturbed calcium handling and lipid signaling [204,205,206]. Cardiac dysfunction further aggravates tissue hypoxia that perpetuates systemic oxidative stress.

COPD-related systemic inflammation, oxidative stress and accelerated cardiovascular aging can directly act on the ventricular muscle and activate signaling pathways leading to maladaptive remodeling and HF [197,207]. In addition, arterial stiffness and hypertension developing in COPD increases left ventricular load and impairs ventriculo-arterial coupling, which also contributes to the development of HF [208]. Accelerated atherosclerosis and endothelial dysfunction increase the occurrence of coronary heart disease (CHD), too. Indeed, approximately 15% of COPD patients also suffer from concomitant CHD [209,210].

## 4. Biomarkers of Oxidative Stress in COPD and Cardiovascular Diseases

### 4.1. Biological Biomarkers

A multitude of studies is available in the literature that addressed characterize systemic and local oxidative stress in association with COPD and various forms of cardiovascular diseases [1,2,211,212,213]. In addition, several biomarkers of oxidative stress are available in the blood, tissues, and other biological samples, such as exhaled breath condensate and sputum [1,211,214]. However, the direct measurement of ROS production is challenging because of the short half-life of reactive oxidants. Therefore, it is more feasible to assess oxidative stress by measuring oxidation target products, such as lipid peroxidation end products and oxidized proteins, as well as the activities of enzymes of the oxidant and antioxidant pathways [215].

Regarding COPD, circulating biomarkers have been widely assessed to correlate with disease and disease severity. These studies relate the systemic manifestation of oxidative stress to COPD rather than local oxidative stress of the lungs. However, samples obtained directly from the respiratory system, such as exhaled breath condensate and sputum, are more informative about the local oxidative burden [1,211]. Table 1 summarizes the biological samples and biomarkers used for evaluating oxidative stress in COPD. Among these, the measurement of a lipid peroxidation product, malondialdehyde (MDA) level, and its reaction with thiobutyric acid to obtain thiobutyric acid reactive substances (TBARS) is the most frequently applied approach to assess oxidative damage. The elevation of MDA in COPD is the most consistent finding among studies which relate oxidative stress to COPD [76,83,86,89,90,216,217,218,219,220,221,222,223,224,225,226,227,228,229,230]. Measurement of protein and non-protein thiols in various biological samples is also an comprehensive tool to evaluate ROS activity. Thiols undergo oxidation in the presence of ROS, constituting an essential component of the intra- and extracellular antioxidant defense system. Therefore, the level and ratio of reduced and oxidized thiols can characterize the oxidative state of the body. In COPD, glutathione (GSH) and its oxidized products are widely used markers of oxidative stress (Table 1) [1,213]. Assessment of antioxidant pathways in COPD has been undertaken by measuring the total antioxidant capacity and enzymatic antioxidant activity of SOD, CAT and GPx. Most studies found decreased antioxidant activity, especially when circulating markers were measured [83,86,88,89,90,216,228,230,231,232,233]. However, higher CAT and SOD activity in sputum was found in exacerbated COPD, most probably due to compensatory response during infectious inflammation [49]. In addition, protein oxidation products, lipid peroxidation products of membrane lipids and phospholipids (hexanal, heptanal, nonanal, acrolein, 8-isoprostane), as well as markers of inflammatory processes induced by oxidative stress, such as leukotrienes can also be used to characterize oxidative burden in COPD (for selected studies see Table 1) [1,213].

Oxidative stress in cardiovascular diseases can also be assessed by measurement of circulating blood biomarkers similar to COPD. In addition, the measurement of fluorescent oxidation products (FlOPs), as a stable biomarker of global oxidative damage reflecting oxidation of lipids, proteins, DNA, and carbohydrates, has been used to assess oxidative stress in various CVDs [234,235,236,237], and may also be of growing interest in respiratory disorders [238,239]. However, the evaluation of local oxidative stress in the heart and vasculature has limited relevance due to the limited availability of tissue samples. The wide literature on oxidative stress in cardiovascular diseases (including reports on human and animal studies) also shows increased oxidant and decreased antioxidant activity in various disease conditions, including hypertension, atherosclerosis, vascular aging, ischemic heart, and cerebral diseases [39,55,57,74,75,77,212]. Interestingly, in atherosclerotic conditions, several studies have shown increased antioxidant activity using blood markers, which may show the compensatory upregulation of antioxidant defense mechanisms in this condition [240,241,242]. The findings of selected representative studies are summarized in Table 2.

### 4.2. Heart Rate Variability—A Potential Non-Conventional Biomarker of Oxidative Stress in COPD and CVD

Impaired autonomic control is a shared characteristic of COPD and cardiovascular diseases and is also associated with inflammation and oxidative stress [243,244,245]. The strong association between bronchial and cardiac vagal tone is also established in the literature [246]. Autonomic dysfunction can be detected by alterations in heart rate variability (HRV). HRV describes the fluctuation in the time interval between heartbeats brought about by oscillating regulatory mechanisms which affect heart rate mainly by modifying the balance of sympathetic and parasympathetic effects on the heart. Numerous parameters—time-domain, frequency-domain and non-linear HRV indices, can be used to characterize the HRV complexly. These parameters are calculated by defining interbeat intervals from continuous ECG recordings obtained over a specified period (2 min to 24 h). High HRV generally represents better body resilience to physiological and pathological challenges and is associated with better health and cardiovascular status [247,248].

In COPD, decreased HRV has been detected in several studies. Moreover, depressed HRV is related to the risk of exacerbations [249,250,251]. Although cardiovascular diseases are also associated with decreased HRV, alterations of certain HRV indices have been proposed to be applicable for assessing prognosis in post-infarction patients and in patients with congestive heart failure [252,253,254,255]. Not surprisingly, several studies also found a correlation between HRV depression and oxidative stress [256,257,258]. These observations suggest that HRV parameters could be used as a non-invasive biomarker of oxidative stress in COPD and CVDs. However, this requires further extensive research. The rationale for the idea is that parameters similar to HRV indices can be obtained from peripheral arterial pulse wave recordings, which are extensively available for analysis, as a wide variety of smart wearable accessories are equipped with photoplethysmographic detectors capable of capturing pulse wave signals [259].

**Table 1 antioxidants-12-01196-t001:** Biomarkers of systemic and local oxidative stress in COPD. Representative studies reporting the association of oxidative stress biomarkers in various biological samples with COPD. Abbreviations: GSH—reduced glutathione, SOD—superoxide dismutase, CAT—catalase, GPx—glutathion peroxidase, MDA—malondialdehyde, AOPP—advanced oxidation protein products, LTB4—leukotriene B4. ↓: decrease in level/activity; →: unchanged level; ↑: increased level/activity.

Sample	Biomarker	Finding	Reference
**Blood (systemic oxidative stress)**			
*erythrocytes*	reduced GSH	↓ in COPD patients (n = 236) vs controls (n = 150) and correlates with disease severity—all patients are smokers or ex-smokers	[216]
		↓ in stable COPD patients (n = 41) vs. controls (n = 30); and further decreased in exacerbated COPD (n = 21)—varying smoking status	[218]
	SOD activity	↓ in COPD patients (n = 140) vs. healthy controls (n = 75)—varying smoking status	[83]
		↓ in COPD patients (n = 234) vs. healthy controls (n = 182)—varying smoking status	[233]
		↓ in COPD patients (n = 82) vs. non-smoking healthy controls (n = 22)	[86]
		↓ in stable COPD patients (n = 21) vs. non-smoking healthy controls (n = 24)	[88]
	CAT activity	↓ in COPD patients (n = 236) vs controls (n = 150) and correlates with disease severity—all patients are smokers or ex-smokers	[216]
		↓ in COPD patients (n = 140) vs. healthy controls (n = 75)—varying smoking status	[83]
		→ comparable in COPD patients (n = 82) and non-smoking healthy controls (n = 22)	[86]
	GPx activity	↓ in COPD patients (n = 236) vs. controls (n = 150)—all patients are smokers or ex-smokers	[216]
		↓ in COPD patients (n = 140) vs. healthy controls (n = 75)—varying smoking status	[83]
		↓ in COPD patients (n = 82) vs. non-smoking healthy controls (n = 22)	[86]
		↓ in COPD patients (n = 20) vs. healthy controls (n = 50)—varying smoking status	[232]
*plasma*	MDA	↑ in COPD patients (n = 236) vs. controls (n = 150)—and correlates with disease severity. All patients are smokers or ex-smokers	[216]
		↑ in stable COPD patients (n = 41) vs. controls (n = 30); and further decreased in exacerbated COPD (n = 21)—varying smoking status	[218]
		↑ in COPD patients (n = 140) vs. healthy controls (n = 75)—varying smoking status	[83]
		↑ in COPD patients (n = 82) vs. non-smoking healthy controls (n = 22)	[86]
		↑ in COPD patients (n = 20) vs. healthy controls (n = 50)—varying smoking status	[232]
		↑ in COPD patients (n = 100) vs. controls (n = 100)—varying smoking status	[221]
		↑ in COPD patients (n = 100) vs. controls (n = 100)—varying smoking status	[222]
		↑ in healthy smokers (n = 30) and in patients with stable (n = 7) and exacerbated COPD (n = 31) than in healthy non-smokers (n = 30)	[223]
		↑ in COPD patients (n = 106) vs. controls (n = 45)—varying smoking status	[225]
		↑ in COPD patients exposed to wood smoke (n = 30) and tobacco smoking (n = 30) vs. healthy controls (n = 30)	[226]
		↑ in COPD patients (n = 815) vs. controls (n = 530)—varying smoking status—meta-analysis	[227]
		↑ in severe COPD patients (n = 74) vs. controls (n = 41)—varying smoking status	[228]
		↑ in COPD patients (n = 26) vs. controls (n = 28) –smoking status n.a.	[229]
		↑ in smoker COPD patients (n = 202) vs. smoker controls without COPD (n = 136)	[89,230]
		↑ in patients with exacerbated (n = 43) and stable (n = 35), and in healthy smokers (n = 14) vs. healthy non-smokers (n = 14)	[90]
		→ comparable in ex-smoker COPD patients (n = 11) and non-smoking healthy controls (n = 12), exercise induces increase only in COPD	[260]
	AOPP	↑ in severe COPD patients (n = 74) vs. controls (n = 41)—varying smoking status	[228]
	reduced GSH	↓ in COPD patients (n = 20) vs. healthy controls (n = 50)—varying smoking status	[232]
		↓ in chronic smokers with stable COPD (n = 20) and without COPD (n = 20) vs. healthy non-smokers (n = 20)	[261]
		↓ in smoker COPD patients (n = 202) vs. smoker controls without COPD (n = 136)	[89,230]
		↓ in patients with exacerbated (n = 43) and stable (n = 35), and in healthy smokers (n = 14) vs. healthy non-smokers (n = 14)	[90]
	SOD activity	↓ in severe COPD patients (n = 74) vs. controls (n = 41)—varying smoking status	[228]
		↓ in patients with exacerbated (n = 43) and stable (n = 35), and in healthy smokers (n = 14) vs. healthy non-smokers (n = 14)	[90]
		↓ in patients with stable COPD (n = 96) vs. controls without COPD (n = 96)—varying smoking status	[231]
	CAT activity	↓ in smoker COPD patients (n = 202) vs. smoker controls without COPD (n = 136)	[89,230]
		→ comparable in patients with stable COPD (n = 96) and without COPD (n = 96)—varying smoking status	[231]
	GPx activity	↓ in smoker COPD patients (n = 202) vs. smoker controls without COPD (n = 136)	[89,230]
		↓ in patients with exacerbated (n = 43) and stable (n = 35), and in healthy smokers (n = 14) vs. healthy non-smokers (n = 14)	[90]
		↓ in COPD patients (n = 82) vs. non-smoking healthy controls (n = 22)	[86]
*whole blood*	total glutathione	↑ in COPD patients (n = 140) vs. healthy controls (n = 75)—varying smoking status	[83,86]
		↑ in COPD patients (n = 82) vs. non-smoking healthy controls (n = 22)	[86]
	GPx activity	↓ in stable COPD patients (n = 21) vs. non-smoking healthy controls (n = 24)	[88]
**Exhaled air (systemic/local oxidative stress)**			
	CO	↑ in ex-smokers with COPD (n = 15) and in smokers with COPD (n = 15) vs. non-smoking healthy controls (n = 10)	[262]
	ethane	↑ COPD (n = 12) vs. healthy (n = 14) (all ex-smokers)	[263]
**Exhaled breath condensate (systemic/local oxidative stress)**			
	hexanal, heptanal, nonanal	↑ in patients with stable COPD (n = 20) vs. non-smoking healthy subjects (n = 20), but not vs. smoking controls (n = 12)	[220]
		↑ in patients with COPD (n = 11; smokers and ex-smokers) vs. non-smoking controls (n = 9)	[219]
	MDA	↑ in patients with stable COPD (n = 20) vs. non-smoking healthy subjects (n = 20), and also vs. smoking controls (n = 12)	[220]
		↑ in patients with COPD (n = 11; smokers and ex-smokers) vs. non-smoking controls (n = 9)	[219]
		↑ in patients with COPD (n = 73) vs. healthy non-smokers (n = 14); an inverse correlation between MDA concentrations and FEV1(%) was found	[217]
		→ comparable values in patients with exacerbated COPD (n = 34), stable COPD (n = 21) and healthy controls (n = 20)—all ex-smokers	[76]
		↑ in patients with COPD (n = 53) vs. healthy (n = 10); MDA correlates with disease severity—all patients were retired coal miners with varying smoking status	[224]
	H_2_O_2_	↑ in patients with COPD (n = 30) vs. healthy (n = 10) and increases with disease severity—all smokers	[264]
		↑ in patients with stable COPD (n = 12) and with exacerbated COPD (n = 19) (smokers and ex-smokers) vs. healthy never-smokers (n = 10)	[265]
	pH	↓ in COPD exacerbation vs. recovery (n = 29)—current and ex-smokers	[266]
		Condensate pH remained unchanged during COPD exacerbation, both in smokers (n = 21) and ex-smokers (n = 17)	[267]
	nitrotyrosine	↑ in patients with COPD (n = 53) vs. healthy (n = 10)—patients were retired coalminers with varying smoking status	[224]
	8-isoprotane	↑ in exacerbating COPD patients (n = 21) and fell after treatment with antibiotics	[268]
		↑ in patients with COPD (n = 30) vs. healthy (n = 10)—all smokers	[264]
	LTB4	↑ in exacerbating COPD patients (n = 21) and fell after treatment with antibiotics	[268]
		↑ in steroid naïve (n = 20) and steroid treated patients with COPD (n = 25) compared to control subjects (n = 15)—all ex-smokers	[269]
**Sputum (local oxidative stress)**			
	hexanal, heptanal, nonanal	↑ in patients with COPD (n = 11; smokers and ex-smokers) vs. non-smoking controls (n = 9)	[219]
	MDA	↑ in patients with stable COPD (n = 21) vs. healthy controls (n = 20); increased further iv exacerbated COPD patients and decreased during recovery (n = 34)—all ex-smokers	[76]
		↑ in patients with COPD (n = 11; smokers and ex-smokers) vs. non-smoking controls (n = 9)	[219]
	SOD	SOD activity was comparable between stable COPD patients and (n = 24) and healthy controls (n = 23); but it increased in COPD exacerbation (n = 36)—all patients were ex-smokers	[49]
	CAT	CAT activity was comparable between stable COPD patients and (n = 24) and healthy controls (n = 23); but it increased in COPD exacerbation (n = 36)—all patients were ex-smokers	[49]

**Table 2 antioxidants-12-01196-t002:** Circulating biomarkers in cardiovascular diseases. Selected studies show the association between blood biomarkers of oxidative stress and various cardiovascular disease conditions. Abbreviations: CV—cardiovascular, GSH—reduced glutathione, CAD—coronary artery disease, SOD—superoxide dismutase, BMI—body mass index, IHD—ischemic heart disease, CAT—catalase, GPx—glutathione peroxidase, ox-LDL—oxidized low-density lipoprotein, TIA—transient ischemic attack, FlOPs—fluorescent oxidation products, CHD—coronary heart disease. ↓: decrease in level/activity; ↑: increased level/activity.

Biomarker	CV Disease	Finding	Reference
Reduced GSH	Atherosclerosis, arterial aging	lower GSH is a predictor of intima/media thickness	[270,271]
	Hypertension	↑ GSH increased glutathione-related antioxidant defense in treated hypertensives	[272]
	CAD	↓ in angiographically proven CAD	[240]
SOD activity	Arterial aging	negatively correlated with systolic and diastolic blood pressure, low serum SOD activity is an independent predictor of carotid intima/media thickening	[273]
	Hypertension	↓ in hypertensive patients regardless of BMI	[274]
	IHD, CAD	↑ in angiographically proven CAD and IHD	[240,241,242]
CAT activity	Hypertension	↓ in hypertensive patients regardless of BMI	[274]
	IHD	↑ in men with IHD	[242]
GPx activity	Atherosclerosis	↓ in prevalent atherosclerosis and lower values are associated with an increased risk of future cardiovascular events	[275]
	Hypertension	lower levels associated with high blood pressure in black women	[276]
	IHD	↓ in men with IHD	[242]
	any cardiovascular events	lower GPx is associated with a higher risk of CV events	[277]
MDA	Atherosclerosis, arterial aging	↑ with carotid intima/media thickening	[271]
	Hypertension	↑ in untreated hypertension	[278,279]
	CAD	↑ in angiographically proven CAD	[240]
ox-LDL	Atherosclerosis, arterial aging	↑ associated with carotid intima/media thickening, and higher arterial stiffness	[271,280]
	Hypertension	↑ in hypertensive men and prehypertensive subjects of both genders	[281,282]
	CAD	↑ ox-LDL associated with CAD, with the severity of CAD and was found to be prognostic for CAD events	[283,284,285,286]
	Stroke	higher values are associated with cerebrovascular events and increased risk of recurrent stroke in TIA patients	[287,288,289]
FlOPS	CHD	an independent predictor of CHD events in men	[236]
		higher levels associated with the risk of CHD in women	[235]

## 5. Conclusions and Future Perspectives

The pathogenesis of COPD and its most frequent cardiovascular comorbidities is linked via shared genetic, environmental and lifestyle risk factors and numerous pathophysiological processes, including systemic inflammation, endothelial dysfunction, and accelerated aging. Many of these are strongly related to oxidative stress in a complex manner. On the one hand, they are activated by exogenous and endogenous oxidative radicals. On the other, they impose a further oxidative burden on the body by inducing ROS production and weakening antioxidant defense mechanisms. As oxidative stress is a common mechanism driving and perpetuating COPD and coexisting CVD progression that can be monitored successfully by several biological and other potential physiological biomarkers, therapeutic approaches to restore oxidative balance have been the focus of extensive research in the last few decades. Strategies to influence oxidative balance with dietary supplementation and drugs targeted at different oxidative stress pathways have been extensively reviewed recently [2,290]. Though there are promising observations with dietary supplementation of antioxidants such as vitamin C, vitamin E, resveratrol and flavonoids and with the application of thiol-based antioxidants, such as N-acetylcysteine and carbocysteine, the exact place of these treatments in COPD and CVD prevention and therapy is still not established [2,291]. There are also attempts to normalize oxidative balance with antioxidant mimetics (SOD, catalase, GPx), NOX and MPO inhibitors, and Nrf2 activators, but their application is in the phase of preclinical and clinical studies [2]. The antioxidant capacity of the body can also be influenced positively by supporting anti-aging processes. Indeed, activation of SIRTs with NAD^+^ precursor supplementation has been shown to benefit the respiratory and cardiovascular systems [292,293,294,295,296]. Also, there is evidence to show the potential benefit of Klotho treatment/supplementation [297,298]. For completeness, physical activity and pulmonary rehabilitation should not be excluded from the possible therapeutic approaches to restore redox status. Exercise enhances antioxidant response, decreases age-related oxidative stress, improves endothelial function, and reduces inflammatory and oxidative signaling, thereby protecting cardiovascular health [299,300]. Pulmonary rehabilitation has also benefited redox responses in COPD patients [301,302,303]. As restoration of oxidative balance is a preventive/therapeutic approach which could favorably influence the underlying processes driving COPD and CVD development, studies to understand better signaling pathways that orchestrate the derangement of oxidative-antioxidative balance are essential to establish antioxidant therapy in COPD patients.

## Figures and Tables

**Figure 1 antioxidants-12-01196-f001:**
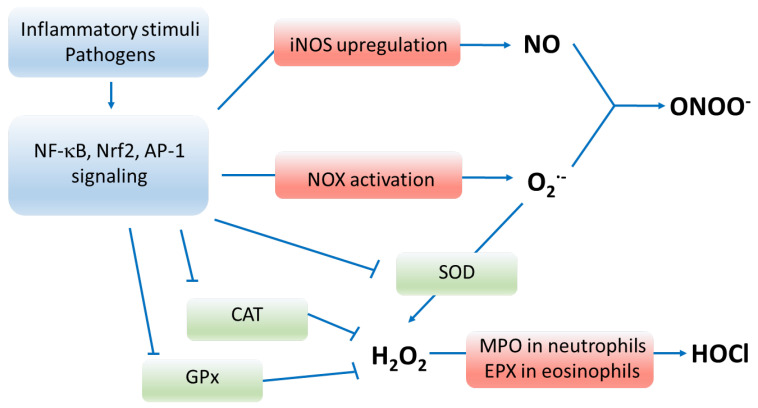
Pathways of oxidative stress. Enzymes marked in red participate in producing oxygen radicals, whereas enzymes marked in green deactivate reactive oxygen species. Inflammatory stimuli, pathogens and oxidants upregulate and activate signaling via NF-κB, Nrf2 and AP-1 transcription factors that result in enhanced production of reactive species and depressed functioning of antioxidant enzymes. Abbreviations: NF-κB—nuclear factor kappa-light-chain-enhancer of activated B cells; Nrf2—nuclear factor erythroid 2-related factor 2; AP-1—activator protein 1; iNOS—inducible nitric oxide synthase; NO—nitric oxide; NOX—nicotinamide adenine dinucleotide phosphate oxidase; O_2_^•−^—superoxide anion, ONOO^−^—peroxynitrite; SOD—superoxide dismutase; CAT—catalase; GPx—glutathione peroxidase; H_2_O_2_—hydrogen peroxide; MPO—myeloperoxidase; EPX—eosinophil peroxidase; HOCl—hypochlorous acid; RNS—reactive nitrogen species; RCS—reactive carbonyl species.

**Figure 2 antioxidants-12-01196-f002:**
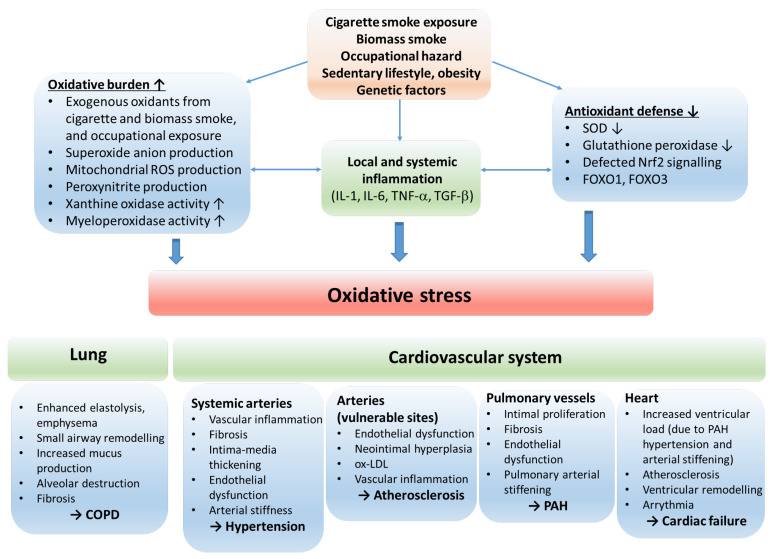
The role of oxidative stress in the etiology of COPD and cardiovascular comorbidities. The oxidative balance of the body is disturbed by risk factors resulting in inflammation, increased oxidative burden and production of reactive oxygen radicals, and reduction in antioxidant defense mechanisms. The consequential oxidative stress stimulates processes that lead to COPD and cardiovascular disorders. Abbreviations: ROS—reactive oxygen species; IL—interleukin; TNF—tumor necrosis factor; SOD—superoxide dismutase; Nrf2—nuclear factor erythroid 2-related factor 2; FOXO1, FOXO3—forkhead box O1 and O3; COPD—chronic obstructive pulmonary disease; ox-LDL—oxidized low-density lipoprotein; PAH—pulmonary arterial hypertension.

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
