# Peer review of "The Role of Oxidative Stress and Antioxidants in Cardiovascular Comorbidities in COPD"

_antioxidants, 2023, doi:10.3390/antiox12061196_

Round 1
Reviewer 1 Report
To the authors:
Overall comments
In the present paper, the authors reviewed the role of oxidative stress and antioxidants in cardiovascular comorbidities in COPD by describing the oxidative stress pathways, and explaining how oxidative stress may be the link between COPD and cardiovascular comorbidities. They described the various biomarkers of systemic and local oxidative stress in COPD and in cardiovascular diseases. Finally, they gave future perspectives in the field.
The research question is of interest, and was scarcely investigated in the literature. To the best of my knowledge, the last similar paper was published in 2016. The present review extended and completed the previous one. The paper is well-written and easy to read.
See below my comments along the text.
Abstract. Line 12: contributes instead of contribute.
1. Introduction. I suggest to cite the previous similar review in the field. To the best of my knowledge: https://www.ncbi.nlm.nih.gov/pmc/articles/PMC4876483/. But others may exist. I also suggest to clearly state that this review update and extend the previous one(s) by describing the biomarkers, by going beyond stroke events …
2. Pathways of oxidative stress. I am wondering of the mirror game between the text of parts 2.1 and 2.2 and Figure 1. Figure 1 is very simple and easy to read, but reflects a complex interplay between ROS and RNS production, antioxidant defenses, various pathways and nuclear factors (not only NF-kB, but also Nrf2 and AP1). I understood the effort of the authors to simplify the purpose but I ask them to reconsider Figure 1 by including at least neutrophils (and why not eosinophils?), oxidative and nitrosative pathways and the three main nuclear factors. Then, Figure 1 will perfectly introduce and describe the oxidative burden and the antioxidant defense parts of Figure 2.
Line 99. I suggest to move here lines 202 to introduce Nrf2. If needed, AP1 should also be introduce here.
Line 104. Exogenous radicals may also come from occupational exposures that is one of the main risk factor of COPD. I suggest to add this information here and in Figure 2, along with a reference as this risk factor in the text. For example: https://pubmed.ncbi.nlm.nih.gov/35427530/.
3. Oxidative stress – a link between COPD and cardiovascular comorbidities. Line 157: Please spell in full FVC and FEV1, and explain what they are useful to. Line 256: Nrf2 instead of NRF2?
4. Biomarkers of oxidative stress in COPD and cardiovascular diseases. Lines 489-498 and Table 2. I strongly suggest to add the Fluorescent Oxidation products (FlOPS). FlOPs reflect a global measurement of oxidation of lipids, proteins, carbohydrates and DNA [1], and is of growing interest for epidemiological studies [2], as an easily quantifiable and stable biomarker of damage due to oxidative stress. Plasma FlOPs level was found to be associated with chronic diseases such as coronary heart disease (CHD) (e.g. incidence of CHD among men without previous cardiovascular events, and risk of future CHD in healthy women) [3,4].
References
1. Dillard, C.J.; Tappel, A.L. Fluorescent Damage Products of Lipid Peroxidation. Methods Enzymol 1984, 105, 337–341, doi:10.1016/s0076-6879(84)05044-8.
2. Wu, T.; Willett, W.C.; Rifai, N.; Rimm, E.B. Plasma Fluorescent Oxidation Products as Potential Markers of Oxidative Stress for Epidemiologic Studies. Am. J. Epidemiol. 2007, 166, 552–560, doi:10.1093/aje/kwm119.
3. Wu, T.; Rifai, N.; Willett, W.C.; Rimm, E.B. Plasma Fluorescent Oxidation Products: Independent Predictors of Coronary Heart Disease in Men. Am J Epidemiol 2007, 166, 544–551, doi:10.1093/aje/kwm120.
4. Jensen, M.K.; Wang, Y.; Rimm, E.B.; Townsend, M.K.; Willett, W.; Wu, T. Fluorescent Oxidation Products and Risk of Coronary Heart Disease: A Prospective Study in Women. J Am Heart Assoc 2013, 2, e000195, doi:10.1161/JAHA.113.000195.
Author Response
Dera Reviewr,
Thank you very much for your comments that were very useful to further develop our manuscript. Please, find our point by point response in the attached document.
Looking forward for your response,
Best regards
Ildiko Horvath
corresponding author

Reviewer 2 Report
The review by Miklos and Horvath is well written and comprehensive, and includes an impressive amount of references. I only have a few comments/suggestions.
1. In the section relevant to oxygen radicals and vascular ageing, please consider to include a mention to arginase, an enzyme known to reduce L-arginine and nitric oxide availability, and increase peroxynitrate thus contributing to airway hypercontractility and vascular remodeling.
2. In the section on accelerated ageing, besides sirtuins and Klotho, the Authors might want to discuss also leukocytes telomere shortening and DNA damage in terms of epigenetic modifications induced by oxidative stress and inflammation in COPD.
3. Among the therapeutic approaches to restore the redox status in COPD please consider mentioning also the effects of pulmonary rehabilitation and exercise training.
4. Please verify acronyms definition and consistency throughout the manuscript. For example NO is not defined at first appearance and is written both as .NO and NO. Please modify where necessary. Thank you.
Author Response
Dear Reviewer,
Than you very much for your comments calling our attention to some important areas related to the topic and develop our manuscript. Attached, please, find our point by point responses.
Looking forward for your response.
Best regards
Ildiko Horvath
corresponding author
